# HTS-Adapt: A Hybrid Training Strategy with Adaptive Search Region Adjustment for MILPs

## Abstract

Mixed Integer Linear Programming (MILP) problems are essential for optimizing complex systems but are NP-hard, posing significant challenges as the problem scale and complexity increase. Recent advances have integrated machine learning to predict partial solutions by exploiting structural patterns in MILP instances. However, existing methods often suffer from inaccurate and infeasible predictions, limiting their practical utility. In this work, we improve the Contrastive Predict-and-Search (ConPaS) framework by introducing a Hybrid Training Strategy with Adaptive Search Region Adjustment mechanism (HTS-Adapt). HTS selectively applies label-based learning and contrastive learning based on the structural properties of variables, improving prediction accuracy. Adapt dynamically adjusts the search space to mitigate infeasible predictions, thereby reducing computational overhead. Experiments demonstrate that our approach achieves a notable performance enhancement by improving prediction accuracy and reducing the search space, proving its effectiveness in addressing real-world MILP challenges. Compared to the MILP solver SCIP, our method achieves an average reduction of more than 50% in the solution gap across four MILP datasets.

## 1 Introduction

Mixed Integer Linear Programming (MILP) is a widely used framework for decision-making in optimizing complex systems, such as manufacturing production lines (Pochet & Wolsey, 2006) and global supply chains (Sharifzadeh et al., 2015). Its expressive power allows practitioners to model complex decision-making tasks under combinatorial constraints. However, MILP problems are inherently NP-hard, as problem size or structural complexity increases, even state-of-the-art solvers struggle to deliver solutions efficiently (Lodi & Zarpellon, 2017). MILP instances often exhibit learnable patterns and structural similarities that can be effectively leveraged by machine learning techniques. This has prompted recent progress in solving MILP problems by integrating machine learning into classical optimization frameworks to enhance the performance and scalability of traditional solvers (Gasse et al., 2019). Broadly, existing approaches fall into two categories. The first line of work focuses on enhancing the traditional branch-and-bound (BnB) algorithm (Land & Doig, 2010) by learning to make key decisions such as branching variable selection (Gasse et al., 2019), node selection (He et al., 2014b), and cutting (Huang et al., 2022), typically by training models on historical solving patterns. The second line of research, which is the focus of this paper, follows an end-to-end paradigm that directly predicts high-quality solutions using machine learning (Nair et al., 2020). These methods predict assignments for a subset of variables to reduce the complexity of the problem and guide the search process. For instance, Neural Diving (ND) (Nair et al., 2020) adopts a predict-then-fix strategy, where predicted variable assignments are fixed to simplify the problem. However, such prediction-based methods are highly sensitive to errors, as fixing incorrect variables can result in suboptimal or even infeasible solutions. The Predict-and-Search (PaS) (Han et al., 2023) framework addresses this limitation by using a trust-region strategy, thereby improving fault tolerance and solution quality. Further advancements, such as Contrastive Predict-and-Search (ConPaS) (Huang et al., 2024), leverage contrastive learning to enhance the accuracy of model predictions, enabling more reliable guidance in solving MILP.

Despite these advances, existing methods still face critical limitations. Inaccurate predictions often lead to suboptimal or infeasible solutions, a challenge that is difficult to overcome due to the inherent diversity and complexity of MILP datasets. Moreover, existing approaches lack an effective means

to guarantee the feasibility of the predicted solutions. The PaS framework traditionally relies on fixed, predefined region hyperparameters. When extending Solution Prediction and PaS framework to real-world scenarios, the inherent diversity and complexity of real-world datasets often introduce edge cases that fall outside these predefined bounds, rendering them infeasible. This often requires enlarging the search space, which compromises the quality of predicted solutions and increases computational overhead.

In this work, we propose a twofold enhancements to the ConPaS framework. The primary goal of our approach is to improve the prediction accuracy of the model, ensuring that predicted solutions are both high-quality and feasible, while also addressing the challenge of robust generalization in real-world datasets. First, we introduce a Hybrid Training Strategy that integrates label-based learning and contrastive learning. This design is motivated by an empirical observation: Some variables consistently exhibit identical values across high-quality solution sets. For these stable variables, we use label-based learning with cross-entropy loss to directly minimize the discrepancy between predicted probabilities and ground-truth labels. For variables with variation, we apply contrastive learning with InfoNCE loss (Oord et al., 2018) to capture discriminative features among different samples in high-dimensional space. Leveraging the complementary strengths of both approaches, our model achieves more accurate predictions. Second, we propose an Adaptive Search Region Adjustment mechanism to dynamically respond to infeasibility. Unlike traditional approaches that simply expand hyperparameters to mitigate infeasibility, our method utilizes the efficient feasibility assessment capabilities of modern solvers. Upon detecting an instance as infeasible within the current search region, the method promptly expands the search range in a targeted manner. Specifically, by computing the Irreducible Infeasible Subsystem (IIS) (Gleeson & Ryan, 1990), we can precisely identify the variables responsible for the infeasibility and dynamically adjust the search region accordingly. This targeted strategy not only ensures feasibility, but also enables the PaS framework to adapt to diverse real-world MILP instances without sacrificing performance.

These advancements improve the accuracy, efficiency, and adaptability of machine learning-driven MILP solvers, offering a more practical and scalable solution for real-world optimization challenges. Experimental results demonstrate that this enhanced approach significantly increases the number of predicted variables while maintaining accuracy. Built upon SCIP (Maher et al., 2017), our approach consistently outperforms the best baseline by achieving superior solution quality, reducing the average absolute primal gap by more than 50% compared to SCIP.

## 2 RELATED WORK

### 2.1 MACHINE LEARNING FOR EFFICIENT BRANCH-AND-BOUND

The iterative nature of the BnB algorithm (Land & Doig, 2010), combined with the inherent structural properties of MILP problems, such as integer constraints and linear relationships, makes it highly compatible with machine learning (ML) and reinforcement learning (RL) techniques. This synergy has led to substantial improvements in the efficiency of solving MILP problems. Within the BnB framework, variable selection (Gasse et al., 2019; Du et al.; Wei et al., 2025) and node selection (Labassi et al., 2022; Zhang et al.; He et al., 2014a) have emerged as the primary areas of integration for these learning-based approaches, directly influencing the algorithm's search efficiency. Additionally, other critical components of modern MILP solvers, including cutting planes (Paulus et al., 2022; Zhang et al., 2024), presolving (Liu et al., 2024a), and warm-start (Patel, 2024) heuristics, play significant roles in enhancing solver performance. Consequently, these areas have attracted considerable research attention.

### 2.2 MACHINE LEARNING FOR HIGH-QUALITY SOLUTION PREDICTION

In the realm of combinatorial optimization, neural networks can leverage the data distribution of historical optimal solutions to construct approximate mappings, enabling the direct prediction of problem solutions. For MILP problems, Nair et al. (2020) pioneered this approach with ND, where they predict a partial solution for a problem instance and employ SelectiveNet (Geifman & El-Yaniv, 2019) to determine which variables to fix. However, these methods face limitations: fixing incorrect variables can result in suboptimal or even infeasible solutions due to the complex constraint structure of MILPs. To address this limitation and improve robustness, Han et al. (2023) proposed the Predict-

and-Search (PaS) framework, which departs from directly fixing variable assignments. Instead, it employs a trust region strategy to constrain the search space around the predicted solution, thereby enhancing fault tolerance and improving overall solution quality. Similarly, Large Neighborhood Search (LNS) (Liu et al., 2024b) has emerged as a popular framework for predicting high-quality solutions, reflecting a broader trend in this field. Building on these frameworks, the integration of contrastive learning has led to the development of ConPaS (Huang et al., 2024) and CL-LNS (Huang et al., 2023), which further refine prediction accuracy by distinguishing high-quality solutions from suboptimal ones through comparative learning techniques.

## 3 PRELIMINARIES

### 3.1 MIXED INTEGER LINEAR PROGRAMS

Mixed Integer Linear Programming (MILP) refers to linear programming problems that incorporate integer constraints. The MILP problem is typically formulated as:

$$\min_x c^\top x \quad \text{s.t.} \quad Ax \le b, \, l \le x \le u, \, x \in \mathbb{Z}^p \times \mathbb{R}^{n-p} \tag{1}$$

Here, $c$ represents the vector of coefficients in the objective function, while $x$ denotes the decision variable vector. Among these variables, the first $x_1, x_2, \ldots, x_p$ are integer variables, and the remaining $n - p$ variables remain continuous. The matrix $A \in \mathbb{R}^{m \times n}$ contains the coefficients of the linear constraints, and $b \in \mathbb{R}^m$ is a vector representing the right-hand side of these constraints. The condition $l \le x \le u$ defines the lower and upper bounds on the decision variables $x$, where $l$ and $u$ can take values of negative or positive infinity. Since integer variables can be encoded as binary variables, this paper focuses on pure binary programming problems.

### 3.2 BIPARTITE GRAPH REPRESENTATION

Gasse et al. (2019) pioneered the approach of encoding MILP problems as bipartite graph representations, $\mathcal{G} \equiv (\mathcal{W} \cup \mathcal{V}, \mathcal{E})$ where constraints are modeled as constraint nodes $\mathcal{W}$ and variables as variable nodes $\mathcal{V}$, with edges $\mathcal{E}$ connecting these two types of nodes. This innovative mapping transforms an MILP problem into a bipartite graph structure, thereby enabling the application of graph neural networks (GNNs) to leverage the inherent relational structure for enhanced problem solving.

### 3.3 PREDICT-AND-SEARCH

Nair et al. (2020) introduced a definition for the marginal probability of a feasible solution as

$$p(x \mid M) = \frac{\exp(-E(x, M))}{\sum_{x' \in \mathcal{S}_M} \exp(-E(x', M))} \quad E(x, M) = \begin{cases} c^\top x, & x \text{ is feasible}, \\ +\infty, & \text{else}, \end{cases} \tag{2}$$

where $\mathcal{S}_M$ is a set of optimal or near-optimal solutions to $M$, $E(x, M)$ is an energy function of a solution $x$. Extending this concept, the Predict-and-Search (PaS) framework Han et al. (2023) assumes independence among variables of an MILP problem and uses GNNs to model the marginal probability of the solution, expressed as $p_\theta(x|M)$, where $p_\theta(x|M) = (p_\theta(x_1 \mid M), \ldots, p_\theta(x_n \mid M))$. In this formulation, $p_\theta(x_i|M)$ denotes the probability that the variable $x_i$ equals 1. Furthermore, the trust region method was proposed to tackle the challenge of prediction inaccuracies. Specifically, the PaS framework operates by predicting the domain within which the solutions lie, governed by three hyperparameters: $k_0, k_1, \Delta$. This method assigns the $k_1$ variables to value 1 exhibiting the highest marginal probabilities and the $k_0$ variables to value 0 with the lowest marginal probabilities, but allowing a perturbation of size $\Delta$, Here $\Delta$ is called trust-region radius. Let $\delta(k_0, k_1, \Delta) = \left\{ x : \sum_{x_i \in I_{k_0}} x_i + \sum_{x_i \in I_{k_1}} (1 - x_i) \le \Delta \right\}$, a trust region problem emerges as a result:

$$\min_{x \in D \cap \delta(k_0, k_1, \Delta)} c^\top x \tag{3}$$

where $D$ is the feasible region of the original problem.

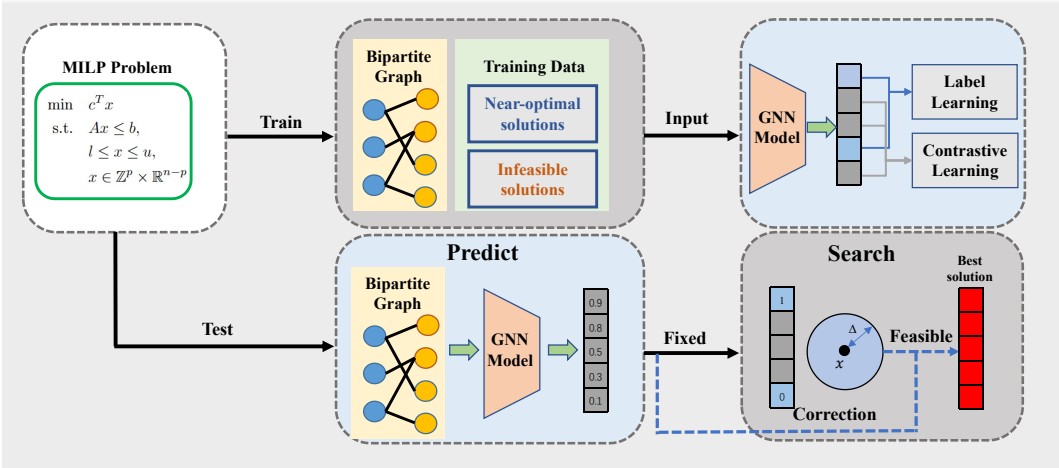

Figure 1: Overview of our proposed framework. During training, we collect near-optimal solutions and infeasible solutions as positive and negative samples, respectively, and route different variables to distinct learning modules based on the characteristics of the near-optimal solutions. At testing, building on the Predict-and-Search (Han et al., 2023) framework, we compute the Irreducible Infeasible Subsystem (IIS) to identify potentially mispredicted variables, thereby adjusting the $\delta$-neighbor for correction.

### 3.4 CONTRASTIVE PREDICT-AND-SEARCH

ConPaS (Han et al., 2023) leverages contrastive learning for MILP solution prediction. It treats high-quality solutions as positive samples and low-quality or infeasible ones as negatives. Using the InfoNCE loss (Oord et al., 2018), the model pulls representations of high-quality solutions closer while repelling those of suboptimal or infeasible solutions, thereby providing more informative guidance for the search process.

## 4 PROPOSED FRAMEWORK

In this section, we introduce our framework-a Hybrid Training Strategy with Adaptive Search Region Adjustment (HTS-Adapt) based on the Contrastive Predict-and-Search (ConPaS) (Huang et al., 2024), with a focus on improving prediction accuracy and adaptability. Our approach introduces two key innovations that work synergistically to address the challenges of prediction reliability and static search regions in the PaS (Han et al., 2023) framework. First, we propose a Hybrid Training Strategy (HTS) that leverages structural insights from high-quality solutions, integrating distinct machine learning techniques. Second, we introduce an Adaptive Search Region Adjustment mechanism (Adapt) that dynamically adjusts the search space by computing the Irreducible Infeasible Subsystem (IIS) (Gleeson & Ryan, 1990), ensuring feasibility while maintaining computational efficiency across a wide range of MILP instances.

### 4.1 HYBRID TRAINING STRATEGY

Our Hybrid Training Strategy (HTS) leverages the structural properties observed in high-quality solutions of mixed integer linear programming (MILP) problems, where certain variables consistently exhibit fixed values while others vary, as illustrated in Appendix B. HTS assigns different learning strategies to different variable types, effectively utilizing the strengths of cross-entropy and contrastive losses, respectively. For stable binary variables, which maintain identical values across a set of high-quality solutions $S_M$ (denoted by index set $I_c$), we apply label-based learning to capture these persistent patterns efficiently, minimizing cross-entropy loss to ensure precision without interference from other variable fluctuations. For binary variables with variation (indexed as $I_{\bar{c}}$), we employ contrastive learning with InfoNCE (Oord et al., 2018) loss to model subtle differences and complex interdependencies, enhancing generalization in various MILP instances. To implement this, we first

collect $\mathcal{S}_M$ optimal or near-optimal solutions, identifying $I_c$ as the set of variables with consistent values and $I_{\bar{c}}$ as those that differ. HTS then designs separate training strategies for $I_c$ and $I_{\bar{c}}$, optimizing the accuracy and robustness of the model. This dual approach combines the precision of label-based learning with the adaptability of contrastive learning, yielding a highly effective training framework for a wide range of problem instances.

The final loss function comprises two components, formulated as:

$$\mathcal{L} = \alpha \cdot \mathcal{L}_{ce} + (1 - \alpha) \cdot \mathcal{L}_{cl}, \tag{4}$$

where $\alpha \in [0, 1]$ is a balancing coefficient that controls the trade-off between the cross-entropy loss $\mathcal{L}_{ce}$ and the contrastive loss $\mathcal{L}_{cl}$. $\mathcal{L}_{ce}$ is the cross-entropy loss for stable variables, defined as:

$$\mathcal{L}_{ce} = -\sum_{i \in I_c} \left[ y_i \log p_\theta(x_i|M) + (1 - y_i) \log(1 - p_\theta(x_i|M)) \right], \tag{5}$$

and $\mathcal{L}_{cl}$ is the InfoNCE loss for variables exhibiting variation, expressed as:

$$\mathcal{L}_{cl} = -\sum_{x_p \in \mathcal{S}_M} \log \frac{\exp(\cos(p_\theta(x|M), x_p, I_{\bar{c}})/\tau)}{\sum_{\tilde{x} \in \mathcal{N}_M \cup \{x_p\}} \exp(\cos(p_\theta(x|M), \tilde{x}, I_{\bar{c}})/\tau)}, \tag{6}$$

Here, $p_\theta(x_i|M)$ denotes the model's predicted probability for variable $x_i$ given the problem instance $M$. For the cross-entropy loss $\mathcal{L}_{ce}$, $y_i$ represents the true label of $x_i$. In the contrastive loss $\mathcal{L}_{cl}$, $\tau$ is the temperature parameter, $\mathcal{N}_M$ is the set of negative samples. Furthermore, we adapted the standard cosine similarity function to enhance the model's discriminative capability, defining it as:

$$\cos(x, y, I) = \frac{x_I \cdot y_I}{\|x_I\|\|y_I\|}, \tag{7}$$

where $x_I$ and $y_I$ are the subvectors of $x$ and $y$ restricted to the indices specified by $I$.

### 4.1.1 POSITIVE SAMPLES COLLECTION

For each MILP instance, we obtain a set of optimal or near-optimal solutions by employing a solver with a fixed runtime $t$, which serves as our positive sample set $\mathcal{S}_M$. In our experiments, we utilize Gurobi (Gurobi Optimization, 2022), a state-of-the-art commercial solver, with a solving time of 3600 seconds, $|\mathcal{S}_M| = 50$.

### 4.1.2 NEGATIVE SAMPLE COLLECTION

We adopt infeasible solutions as negative samples, as proposed by Huang et al. (2024), to streamline the negative sample collection process. Starting with the positive sample set $\mathcal{S}_M$, we first identify the indices of binary variables that remain constant across $\mathcal{S}_M$, denoted as $I_c$. These variables encapsulate patterns inherent to high-quality solutions for the given MILP instance. We then focus on perturbing the binary variables outside this set, indexed as $I_{\bar{c}}$, to enable the model to learn finer-grained features. For each solution $x \in \mathcal{S}_M$, we randomly perturb 10% of the variables in $I_{\bar{c}}$ (i.e., flipping them from 0 to 1 or 1 to 0). If the perturbed solution is infeasible, it is added to the negative sample set $\mathcal{N}_M$. For each $x \in \mathcal{S}_M$, we generate $\beta$ infeasible solutions through this method to form the negative sample set $\mathcal{N}_M$. This approach generates negative samples that better align with HTS.

### 4.2 ADAPTIVE SEARCH REGION ADJUSTMENT

The Adaptive Search Region Adjustment (Adapt) method aims to tackle the challenge of predicted solutions becoming infeasible when applying the Predict-and-Search (PaS) (Han et al., 2023) framework to real-world MILP problems, where the diversity and complexity of datasets inevitably lead to infeasible outcomes. Furthermore, the inherent limitations of neural network predictions also contribute to the infeasibility of predicted solutions. The static strategy adopted in PaS struggles to generalize across diverse instances, as the predefined $\delta$-neighborhood may fail to ensure feasibility or overly expand the search space, thereby compromising the quality of predicted solutions and computational efficiency. To overcome this limitation, our method introduces a dynamic adjustment mechanism that leverages the Irreducible Infeasible Subsystem (IIS) (Gleeson & Ryan, 1990). Specifically, our approach directly identifies the specific variables responsible for inaccurate predictions,

---

**Algorithm 1** Adaptive Search Region Adjustment (Adapt)

---

**Input:** Probability prediction $p_\theta(x|M)$, MILP instance $M$, hyperparameters $k_0$, $k_1$, $\Delta$
**Output:** Feasible solution $x$ (if found)
  1: Initialize unfixed variable set $U \leftarrow \emptyset$, status flag $\gamma \leftarrow$ **false**
  2: Sort variables by descending marginal probability $p_\theta(x_i \mid I)$
  3: **repeat**
  4:    Fix variables not in $U$ using $\delta(k_0, k_1, \Delta)$
  5:    Solve the constrained MILP for up to 1000 seconds
  6:    **if** solution is feasible **then**
  7:       $\gamma \leftarrow$ **true**
  8:       **return** Best feasible solution $x \in \mathbb{R}^n$
  9:    **else**
 10:       Compute IIS (Irreducible Infeasible Subsystem)
 11:       Partition IIS into $\mathcal{C}_1$ and $\mathcal{C}_2$
 12:       $r \leftarrow \frac{|\mathcal{C}_2|}{|\mathcal{C}_1|}$
 13:       **if** $r \leq r_{max}$ **then**
 14:          **for** $v$ in $\mathcal{C}_1$ **do**
 15:             Add $v$ to $U$
 16:          **end for**
 17:       **else**
 18:          **for** $v$ in $\mathcal{C}_1 \cup \mathcal{C}_2$ **do**
 19:             Add $v$ to $U$
 20:          **end for**
 21:       **end if**
 22:    **end if**
 23: **until** $\gamma$ is **true**
 24: **return** Best feasible solution $x$

---

thereby broadening the potential of machine learning-based solution prediction. Consider an MILP problem $M$ with its original constraint set $\mathcal{C}_1$ and an additional constraint set $\mathcal{C}_2$, corresponding to $D$ and $\delta(k_0, k_1, \Delta)$ in Equation (3), respectively. When solving $M$ proves infeasible, a process that typically takes only a few seconds, we compute the Irreducible Infeasible Subsystem (IIS) to pinpoint conflicting constraint sets. This process enables us to isolate the conflicting constraints within $\mathcal{C}_1$ and $\mathcal{C}_2$, thereby identifying the variables causing the conflict. The IIS is defined as a minimal infeasible subset of constraints, characterized by the following properties:

**Infeasibility**: The subset constitutes an infeasible system, meaning no solution satisfies all constraints within it.

**Minimality**: Removing any single constraint from this subset makes the system feasible.

We initially extract constraints belonging to $\mathcal{C}_1$ from IIS and refrain from imposing additional constraints on the variables associated with them. By solving Equation (3) under this adjusted setting, we obtain a feasible solution. Observations indicate that, with suitable choices of $k_0$ and $k_1$, a IIS typically includes only one constraint from $\mathcal{C}_1$. Moreover, due to the sparsity of the MILP coefficient matrix, these constraints generally involve few variables, resulting in a minimal impact on the search region. If the ratio $r = \frac{|\mathcal{C}_2|}{|\mathcal{C}_1|}$, computed within the IIS, exceeds a predefined threshold $r_{\max}$, indicating a high number of inaccurately predicted variables, we extend our approach by also extracting constraints from $\mathcal{C}_2$ to further refine the search region adjustment. As shown in Algorithm 1, the details of the testing phase are presented. Based on Algorithm 1 and the properties of IIS, we derive the following analysis: if the predicted solution violates $k$ constraints in $\mathcal{C}_1$, each IIS computation eliminates at least one of these violated constraints. Consequently, the algorithm requires at most $k$ iterations for the predicted solution to become feasible. Algorithm 2 presents the main steps for computing an IIS. In practice, we compute the IIS using the function provided by Gurobi(see the appendix E).

## 5 EXPERIMENTS

In our evaluation, we focus on three main parts. First, we conduct comparative experiments using SCIP (Maher et al., 2017) across four datasets, benchmarking our method against Predict-and-Search (PaS) (Han et al., 2023) and Contrastive Predict-and-Search (ConPaS) (Huang et al., 2024). We then perform ablation and generalization studies. Due to space limitations, additional results—including experiments with Gurobi (Gurobi Optimization, 2022) as the solver, evaluations on more datasets, and comparisons with extra baselines—are provided in the Appendix.

### 5.1 SETTINGS

#### 5.1.1 BENCHMARK

In our experiments, we utilize four classic combinatorial optimization problems as benchmark tasks: Maximum Independent Set (MIS), Minimum Vertex Cover (MVC) (Garey Michael & Johnson David, 1979), Combinatorial Auction (CA) (Lehmann et al., 2006), and Item Placement (IP) (Gasse et al., 2022). These benchmarks are sourced from Gasse et al. (2019) and ML4CO competition (Gasse et al., 2022). MVC and MIS, both graph optimization problems, are generated using the Barabási–Albert and Erdös-Rényi random graph model (Albert & Barabási, 2002; ERDdS & R&wi, 1959), consisting of 6,000 nodes. The CA instances, representing combinatorial auction problems, are constructed based on the arbitrary relations outlined in Leyton-Brown et al. (2000), comprising 2000 items and 4,000 bids. IP instances are selected from the NeurIPS ML4CO competition. Following the setup of Gasse et al. (2019), we utilize 240 instances for training, 60 instances for validation, and 100 instances for testing.

#### 5.1.2 BASELINES

We evaluate the following methods from existing literature as our baselines, encompassing both learning-based heuristics and an exact optimization solver: PaS, which utilizes predictive models to guide a search algorithm toward high-quality solutions; ConPaS, an extension of PaS that incorporates contrastive learning to improve the predictive model's ability to differentiate between optimal and low-quality solutions; and SCIP (Maher et al., 2017), a open-source solver for mixed integer programming (MIP). The PaS models in our experiments are trained using the code provided by Han et al. (2023). Due to the the code of ConPaS is not available, we implement our own version based on the details from the paper, making adjustments to several parameters to optimize its performance. In our implementation, we use low-quality solutions as negative samples during training.

Table 1: Performance comparison of our approach against baseline methods on benchmarks under a 1,000-second time limit, using the SCIP solver for all methods. The results are averaged over 100 test instances across four problem types. We report the average best objective value-Obj and the absolute primal gap-$\text{gap}_{\text{abs}}$, where $\uparrow$ denotes higher is better, and $\downarrow$ denotes lower is better. Additionally, we show the improvement of our method (Ours+SCIP) over SCIP in terms of $\text{gap}_{\text{abs}}$.

| | CA (BKS 117223.59) | | IP (BKS 5.90) | | MIS (BKS 2628.05) | | MVC (BKS 3480.39) | |
|---|---|---|---|---|---|---|---|---|
| | Obj $\uparrow$ | $\text{gap}_{\text{abs}} \downarrow$ | Obj $\downarrow$ | $\text{gap}_{\text{abs}} \downarrow$ | Obj $\uparrow$ | $\text{gap}_{\text{abs}} \downarrow$ | Obj $\downarrow$ | $\text{gap}_{\text{abs}} \downarrow$ |
| SCIP | 112899.50 | 4324.09 | 13.41 | 7.51 | 2577.60 | 50.45 | 3530.12 | 49.73 |
| PaS+SCIP | 113180.23 | 4043.36 | 12.01 | 6.11 | 2612.42 | 15.63 | 3487.48 | 7.09 |
| ConPaS+SCIP | 113051.31 | 4172.28 | 11.86 | 5.96 | 2613.34 | 14.71 | 3487.26 | 6.87 |
| Ours+SCIP | 113343.56 | 3880.03 | 11.27 | 5.37 | 2618.79 | 9.26 | 3485.70 | 5.31 |
| Improvement | 10.3% | | 28.5% | | 81.6% | | 89.3% | |

#### 5.1.3 METRICS

In our evaluation, we adopt the following metrics to assess all approaches: (1) Primal gap (Berthold, 2006), defined as the normalized difference between the primal bound $v$ and the precomputed best-known solution (BKS) $v^*$, calculated as $\frac{|v-v^*|}{\max(v,v^*,\varepsilon)}$ if $v$ exists and $v \cdot v^* \geq 0$, and set to 1 otherwise. We set $\varepsilon = 10^{-8}$ to avoid division by zero. The BKS $v^*$ is defined as the best objective obtained by

running Gurobi in single-thread mode for 3600 seconds, combined with the best value achieved by all methods within a 1000-second time limit. (2) Absolute primal gap, defined as the absolute difference $|v - v^*|$, denoted by $\text{gap}_{\text{abs}}$.

### 5.1.4 IMPLEMENTATION

Our experiments are conducted on a system with an AMD EPYC 7763 64-Core processor and an NVIDIA GeForce RTX 4090 GPU. SCIP 9.2.2 and Gurobi 11.0.3 are utilized in our experiments. All experiments are conducted in a single-threaded environment. To collect training data, we gather the 50 best solutions for each training instance, using Gurobi with a solving time of 3,600 seconds. During training, we employ the Adam optimizer (Kingma & Ba, 2014) with a learning rate of 0.001, a batch size of 8, and trained for 1,000 epochs with early stopping. For the reproduction of PaS, we utilize the code provided in the original paper. Since ConPaS lacks publicly available code, we made our best effort to implement it, aiming to achieve optimal performance. The partial solution size parameters $(k_0, k_1, \Delta)$ are detailed in the Appendix D and for all experiments we set $\alpha = 0.4$, $\beta = 10$ and $\tau = 0.1$.

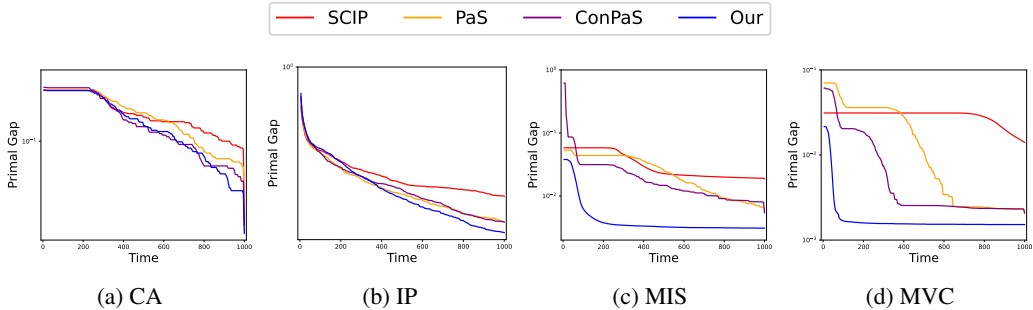

|  |  |  |  |
| :---: | :---: | :---: | :---: |
| (a) CA | (b) IP | (c) MIS | (d) MVC |

Figure 2: We track the primal gap during the solving process using SCIP. All methods—SCIP, PaS, ConPaS, and Ours—are implemented with a 1,000-second time limit, and results are averaged over 100 test instances from four problem types.

### 5.2 RESULTS

Table 1 presents the quality of the final solutions obtained by our method and the baseline approaches within a 1,000-second time limit, reporting the average objective values and absolute primal gaps across 100 test instances. Our method consistently outperforms the baselines across four datasets, almost always achieving the best results in each case. Notably, in the MVC dataset, our approach surpasses the performance of Gurobi with a 3,600-second runtime on most test instances. In the challenging IP dataset, our method achieves a best objective value with an 10.12% improvement in $\text{gap}_{\text{abs}}$ over the best baseline. Similarly, in the MIS dataset, our method demonstrates an 37.05% improvement over ConPaS in terms of $\text{gap}_{\text{abs}}$. In the CA dataset, our method performs comparable to the baselines but still achieves the best results with a slight advantage. While PaS and ConPaS exhibit varying strengths across different datasets, our method consistently delivers superior solution quality. We also present the results on WA(Gasse et al., 2022), another dataset from the ML4CO competition, in Appendix H.

Figure 2 provides a more comprehensive comparison between our method and the baselines in terms of both solving speed and solution quality, illustrating the primal gap as a function of runtime. A rapid decrease in the primal gap curves indicates superior solving performance. Although ConPaS achieves worse solution quality than PaS on certain datasets, it consistently demonstrates faster solving speeds across all datasets. In contrast, our method not only achieves better solution quality but exhibits faster convergence compared to the baselines, showcasing improved performance in both speed and quality across all datasets.

### 5.2.1 GENERALIZATION EXPERIMENTS

To evaluate generalization, we employ the trained models on two benchmark datasets, MIS and MVC, both representative graph optimization problems. We generate 100 larger test instances, each consisting of 8,000 nodes, using the Barabási–Albert and Erdős–Rényi random graph models (Albert & Barabási, 2002; ERDdS & R&wi, 1959). Table 2 presents all test results, including the average best objective value and absolute primal gap. Our method significantly outperforms the baselines on both datasets, nearly always achieving the best objective values.

Table 2: Generalization performance on MIS and MVC datasets with 100 test instances, reporting the average best objective value and absolute primal gap under a 1,000-second time limit. ↑ indicates that higher values are better, while ↓ indicates that lower values are better.

| | MIS (BKS 3493.91) | | MVC (BKS 4641.85) | |
|---|---|---|---|---|
| | Obj ↑ | $\text{gap}_{\text{abs}}$ ↓ | Obj ↓ | $\text{gap}_{\text{abs}}$ ↓ |
| SCIP | 3413.74 | 80.17 | 4786.76 | 144.91 |
| PaS+SCIP | 3477.67 | 16.24 | 4648.31 | 6.46 |
| ConPaS+SCIP | 3483.29 | 10.62 | 4654.73 | 12.88 |
| Ours+SCIP | 3493.82 | 0.09 | 4641.86 | 0.01 |

### 5.2.2 ABLATION STUDY

We conduct an ablation study on the MIS and MVC dataset to evaluate the effectiveness of the HTS and Adapt, using ConPaS-the best performing baseline on this dataset-as the reference for comparison. As shown in Table 3, HTS achieves a 44.9% and 20.8% reduction in $\text{gap}_{\text{abs}}$ compared to ConPaS, highlighting the significant enhancement in prediction accuracy. Similarly, Adapt achieves a 51.5% and 15.1% reduction in $\text{gap}_{\text{abs}}$, underscoring its superior performance in improving prediction accuracy. By integrating HTS and Adapt, our approach achieves the best performance.

Table 3: Ablation study results on the MIS and MVC datasets, comparing ConPaS, HTS-only, Adapt-only, and our approach (Ours) across different settings of $(k_0, k_1, \Delta)$. The setting $(k_0, k_1, \Delta)$ determine whether the Adapt module is enabled, as $\Delta = 0$ may lead to infeasibility for certain instances.

| | MIS | | | MVC | | |
|---|---|---|---|---|---|---|
| Model | $(k_0, k_1, \Delta)$ | Obj ↑ | $\text{gap}_{\text{abs}}$ ↓ | $(k_0, k_1, \Delta)$ | Obj ↓ | $\text{gap}_{\text{abs}}$ ↓ |
| ConPaS | (800, 400, 10) | 2613.34 | 14.71 | (800, 200, 10) | 3487.26 | 6.87 |
| HTS-only | (1200, 600, 1) | 2617.59 | 10.46 | (800, 200, 1) | 3485.83 | 5.44 |
| Adapt-only | (800, 400, 0) | 2618.48 | 9.57 | (800, 200, 0) | 3486.22 | 5.83 |
| Ours | (1200, 600, 0) | 2618.79 | 9.26 | (800, 200, 0) | 3485.70 | 5.31 |

We further perform ablation studies on the weighting parameter $\alpha$, the temperature parameter $\tau$, and the perturbation ratio. Details are provided in Appendix G.

## 6 CONCLUSION AND FUTURE WORK

In this work, we enhanced the ConPaS framework by introducing a Hybrid Training Strategy with Adaptive Search Region Adjustment mechanism (HTS-Adapt), significantly improving prediction accuracy and adaptability for MILP solving. Experimental results demonstrate that our approach outperforms the PaS and ConPaS, with HTS enhancing prediction accuracy and Adapt reducing the search space and computational overhead. For future work, we plan to explore the application of our framework to the more complicated combinatorial optimization problems.

REPRODUCIBILITY STATEMENT

We describe all datasets, model architectures, and training procedures in detail in the main paper. Hyperparameter values are specified in Appendix D, and the code is provided in the supplementary material. Although the code is not publicly released at the time of submission, we will make it available upon acceptance.

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

## A    DETAILS ON THE GNN ARCHITECTURE

In our approach, we represent a MILP instance $M$ as a bipartite graph, a technique adapted from previous works Gasse et al. (2019); Han et al. (2023). The bipartite graph consists of two types of nodes: variable nodes and constraint nodes, with edge and node features inherited from Han et al. (2023).

We employ a Graph Neural Network model similar to that in Han et al. (2023). Initially, node features are processed through a two-layer Multi-Layer Perceptron (MLP), each with 64 hidden units and ReLU activations, mapping the features to a 64-dimensional space ($\mathbb{R}^{64}$). The model then performs two rounds of message passing: the first from the variable nodes to the constraint nodes, and the second from the constraint nodes back to the variable nodes. These message-passing steps utilize graph convolution layers, as described by Gasse et al. (2019), to generate refined variable embeddings. Finally, the embeddings are passed through a second MLP with 64 hidden units per layer and ReLU activation, followed by a sigmoid activation to output a predicted probability vector $p_\theta(x|M)$ for the binary variables.

## B    PREVALENCE OF STABLE VARIABLES IN HIGH-QUALITY MILP SOLUTIONS

Table 4 reports the proportion of stable binary variables (i.e., those fixed across all high-quality solutions $\mathcal{S}_M$) relative to the total number of binary variables in representative MILP instances. A substantial proportion of variables remains fixed in value, motivating our Hybrid Training Strategy (HTS), which applies tailored learning modules to stable and varying variables.

Table 4: Average counts and proportions of stable variables across problem types.

|            | CA    | IP    | MIS   | MVC   |
|------------|-------|-------|-------|-------|
| Count      | 2958  | 399   | 4022  | 2511  |
| Proportion | 74.0% | 38.0% | 67.0% | 41.9% |

This stability arises from inherent structural properties of real-world MILPs. Practical instances exhibit constraint couplings, sparsity, dominance relations, and resource bottlenecks that tightly constrain the feasible region, forcing many variables to adopt identical values across nearly many high-quality solutions. Such consistency is not coincidental but a direct consequence of the problem structure.

This insight underpins HTS: label-based supervision efficiently captures persistent patterns in stable variables, while contrastive learning models the subtle distinctions and intricate dependencies among varying ones.

## C  DETAILS ON THE BENCHMARKS

To provide a comprehensive understanding of the experimental settings used in this work, we present detailed statistical information for all benchmark instances in Table 5. In this appendix, we provide detailed descriptions and MILP formulations for the benchmark problems used in our experiments.

**Minimum Vertex Cover (MVC).**    In the MVC problem, we are given an undirected graph $G = (V, E)$ with a non-negative weight $w_v$ associated with each node $v \in V$. The goal is to select a subset of nodes $V' \subseteq V$ such that every edge in $E$ has at least one endpoint in $V'$, while minimizing the total weight of the selected nodes. The corresponding MILP formulation is as follows:

$$\min \quad \sum_{v \in V} w_v x_v$$
$$\text{s.t.} \quad x_u + x_v \geq 1, \quad \forall (u, v) \in E,$$
$$x_v \in \{0, 1\}, \quad \forall v \in V.$$

**Maximum Independent Set (MIS).**    The MIS problem also operates on an undirected graph $G = (V, E)$, where the objective is to find the largest subset of vertices such that no two selected vertices share an edge. The standard MILP formulation for MIS is given by:

$$\max \quad \sum_{v \in V} x_v$$
$$\text{s.t.} \quad x_u + x_v \leq 1, \quad \forall (u, v) \in E,$$
$$x_v \in \{0, 1\}, \quad \forall v \in V.$$

**Combinatorial Auction (CA).**    In the CA problem, we are given $n$ bids $\{(B_i, p_i)\}_{i=1}^n$ for $m$ items, where each bid $B_i \subseteq [m]$ represents a subset of items and $p_i$ denotes the associated price. The objective is to select a subset of non-overlapping bids—i.e., no item is allocated more than once—in order to maximize total revenue. The MILP formulation is defined as:

$$\max \quad \sum_{i=1}^{n} p_i x_i$$
$$\text{s.t.} \quad \sum_{\substack{i \\ j \in B_i}} x_i \leq 1, \quad \forall j = 1, \ldots, m,$$
$$x_i \in \{0, 1\}, \quad \forall i = 1, \ldots, n.$$

Additional problem descriptions, such as Item Placement and Workload Appointment, can be found on the ML4CO competition website.[1]

Table 5: Statistical information of the benchmarks we used in this paper.

|  | CA | IP | MIS | MVC |
|---|---|---|---|---|
| Constraint Number | 2677 | 195 | 15063 | 29975 |
| Variable Number | 4000 | 1083 | 6000 | 6000 |

## D  DETAILS ON THE HYPERPARAMETERS

Table 6 summarizes the hyperparameters $(k_0, k_1, \Delta)$ used in our method and the baseline approaches (PaS and ConPaS) across all benchmark tasks. For each benchmark, the values are tuned to obtain better solutions. Specifically, for both PaS and ConPaS, we begin by fixing $\Delta$ to 0, 5 , 10, 20, 50 and 100, then varying $k_0$ and $k_1$ from 0% to 50% of the number of binary variables (in 10% increments)

---

[1]https://github.com/ds4dm/ml4co-competition/blob/main/DATA.md

to evaluate their performance on the validation set and determine suitable initial values. Based on these results, we subsequently fine-tuned $\Delta$, $k_0$, and $k_1$ around the identified values to obtain the optimal configuration. Notably, guided by empirical results, our method sets $\Delta = 0$ for MIS and MVC, indicating that a fixed strategy is used. As the CA dataset is intrinsically complex, leading to a substantial number of mispredicted variables, we chose not to activate the Adapt module.

Table 6: Hyperparameters $(k_0, k_1, \Delta)$ used for our method and baselines.

| Benchmark | CA | IP | MIS | MVC |
|---|---|---|---|---|
| PaS | (400,0,100) | (400,5,1) | (1200,600,5) | (800,200,10) |
| ConPaS | (800,0,50) | (400,5,2) | (800,400,10) | (800, 200, 10) |
| Ours | (400,0,100) | (400,5,1) | (1200,600,0) | (800,200,0) |

## E    IRREDUCIBLE INFEASIBLE SUBSYSTEMS

In this study, we utilize the Irreducible Infeasible Subsystem (IIS) to identify variables causing prediction-induced infeasibility and to guide adaptive adjustments of the search region in MILP solving. Theoretically, an IIS is defined as both infeasible and minimal, meaning that removing any single constraint restores feasibility. However, in practice, many solvers, including Gurobi, CPLEX(Cplex et al., 2009), Mindopt(Aliyun Optimization Team, 2025) employ heuristic algorithms like the filtering method (Gleeson & Ryan, 1990; Chinneck, 2007) to identify an IIS, which may result in a non-minimal and non-unique subset. Despite these theoretical limitations, our approach remains effective in practice. The goal is not to exhaustively identify all infeasibility sources or the smallest IIS, but to isolate a sufficiently informative subset of conflicting constraints to detect inaccurate variable predictions. Even a partial IIS offers valuable insights, enabling meaningful adjustments to the search space. In our experiments, we utilized Gurobi's API. This decision was made primarily for proof-of-concept and experimental purposes.

Algorithm 2 provides a simplified IIS computation procedure used in this work. It is included here to clarify how the IIS is conceptually extracted when analyzing infeasibility induced by inaccurate predictions.

To clarify the computational overhead introduced by the IIS module, we provide empirical evidence based on our experiments. As shown in Table 7, the average time for computing IIS typically ranges from 1 to 8 seconds, which accounts for less than 1% of the total solve time per instance.

We also further evaluate the sensitivity of our method to the choice of threshold values $r_{max}$ on the MVC dataset, using SCIP with a time limit of 1000 seconds. Since the number of predicted infeasible instances is relatively small, we report the average objective value only on these cases. The results are summarized in Table 8. From these results, we find that setting $r_{\max}$ excessively large can lead to a slight deterioration in the final objective value. In contrast, choosing a moderately small $r_{\max}$ yields more consistent outcomes, suggesting that a tighter adjustment range provides a good balance between robustness and performance. In our experiments, we set $r_{\max} = 50$.

Table 7: IIS computation time statistics across different datasets. "avg" and "max" denote the average and maximum computation time, and "ratio" represents the percentage of IIS computation time relative to the total solve time.

| Dataset | avg (s) | max (s) | ratio (%) |
|---|---|---|---|
| CA | 1.33 | 1.45 | 0.13 |
| IP | 0.27 | 0.29 | 0.03 |
| MIS | 6.48 | 7.33 | 0.65 |
| MVC | 4.75 | 5.06 | 0.48 |

---

**Algorithm 2** IIS Computation for MILP

---

**Input:** Infeasible MILP instance $M$ with constraints $\mathcal{C}$
**Output:** IIS $\mathcal{I} \subseteq \mathcal{C}$
 1: Initialize IIS candidate $\mathcal{I} \leftarrow \mathcal{C}$
 2: Initialize improved flag $\gamma \leftarrow$ **true**
 3: **while** $\gamma$ is **true do**
 4:    $\gamma \leftarrow$ **false**
 5:   **for** each constraint $c \in \mathcal{I}$ **do**
 6:      Temporarily remove $c$ from $\mathcal{I}$
 7:      Solve the MILP with remaining constraints $\mathcal{I} \setminus \{c\}$
 8:      **if** MILP is feasible **then**
 9:        $c$ is necessary, add it back to $\mathcal{I}$
10:      **else**
11:        $c$ is redundant, permanently remove it
12:        $\gamma \leftarrow$ **true**
13:      **end if**
14:   **end for**
15: **end while**
16: **return** IIS $\mathcal{I}$

---

Table 8: Sensitivity analysis of the threshold parameter $r_{\max}$ on the MVC dataset. A value of $-1$ indicates that only the `else` branch of Algorithm 1 is executed.

| $r_{\max}$ | $-1$ | 40 | 50 | 60 | 100 | 200 |
|---|---|---|---|---|---|---|
| Obj $\downarrow$ | 3486.83 | 3486.83 | 3486.83 | 3486.83 | 3487.17 | 3488.33 |

## F   PERFORMANCE UNDER GUROBI SOLVER

To further demonstrate the effectiveness and robustness of the proposed method, we replicate the experiments using Gurobi(single-threaded) as the underlying solver. The corresponding results are presented below. It can be observed that our method consistently achieves the best performance among all compared approaches across various datasets.

Table 9: Performance comparison of our approach against baseline methods on benchmarks under a 1,000-second time limit, using the Gurobi solver for all methods.

| | CA (BKS 117223.59) | | IP (BKS 5.90) | | MIS (BKS 2628.05) | | MVC (BKS 3480.39) | |
|---|---|---|---|---|---|---|---|---|
| | Obj $\uparrow$ | gap$_{abs}$ $\downarrow$ | Obj $\downarrow$ | gap$_{abs}$ $\downarrow$ | Obj $\uparrow$ | gap$_{abs}$ $\downarrow$ | Obj $\downarrow$ | gap$_{abs}$ $\downarrow$ |
| Gurobi | 116576.63 | 646.96 | 6.65 | 0.75 | 2563.70 | 64.35 | 3514.79 | 34.40 |
| PaS+Gurobi | 116780.64 | 442.95 | 6.60 | 0.70 | 2622.52 | 5.53 | 3481.42 | 1.03 |
| ConPaS+Gurobi | 116665.31 | 558.28 | 6.57 | 0.67 | 2626.27 | 1.78 | 3481.25 | 0.86 |
| Ours+Gurobi | 116812.32 | 411.27 | 6.47 | 0.57 | 2626.96 | 1.09 | 3480.64 | 0.25 |
| Improvement | 36.4% | | 24.0% | | 98.3% | | 99.3% | |

## G   ADDITIONAL ABLATION

The weighting between the two loss components is a critical factor influencing model performance. To investigate which loss contributes more significantly to the improvements, we conduct an additional ablation study. Table 10 reports the results with $\alpha$ varying from 0 to 1 in increments of 0.2. Consistent with the main experiments, we initially used the MIS dataset for evaluation. Furthermore, to validate the effectiveness more comprehensively, we additionally tested on the MVC dataset. The results show that the best performance is achieved at $\alpha = 0.2$, while the worst occurs at $\alpha = 1$ or $\alpha = 0$, where only one module is playing a role. This indicates that both learning modules jointly contribute to the improved model performance. When $\alpha = 0$, the model still performs well, slightly outperforming

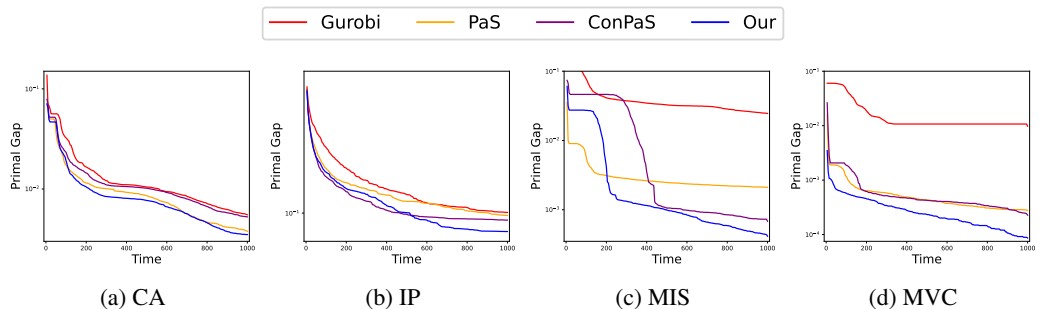

(a) CA     (b) IP     (c) MIS     (d) MVC

Figure 3: Primal gap curves of different methods solved by Gurobi on four problem types, with a 1,000-second time limit.

the pure ConPaS baseline. This suggests that the perturbations applied to variables in $I_{\bar{c}}$ to generate negative samples indeed enable the model to learn more fine-grained information. Notably, the ablation experiments achieve better results on both datasets compared to the main experiments, with $\alpha = 0.2$ yielding superior performance.

The temperature parameter $\tau$ in the contrastive learning component is another crucial factor influencing model performance. We conducted experiments on the IP dataset with different values of $\tau$, and the results are summarized in Table 11. It can be observed that the best performance is achieved when $\tau = 0.1$.

Furthermore, we investigated the effect of the perturbation ratio, with the results reported in Table 12. On the IP dataset, the model achieves its best performance when the ratio is set to $0.15$, yielding a remarkable $17.6\%$ improvement in $\text{gap}_{abs}$ compared to ConPaS.

These ablation studies demonstrate that our proposed method still holds substantial potential for further improvement.

Table 10: Effect of the loss weight $\alpha$ on model performance, using SCIP with a 1,000-second time limit.

|  | MIS | | MVC | |
|---|---|---|---|---|
| $\alpha$ | Obj $\uparrow$ | $\text{gap}_{abs} \downarrow$ | Obj $\downarrow$ | $\text{gap}_{abs} \downarrow$ |
| 0.0 | 2618.50 | 8.38 | 3486.21 | 5.82 |
| 0.2 | **2619.20** | **7.68** | **3485.14** | **4.75** |
| 0.4 | 2618.99 | 7.89 | 3485.70 | 5.31 |
| 0.6 | 2618.76 | 8.12 | 3485.23 | 4.84 |
| 0.8 | 2618.54 | 8.34 | 3485.43 | 5.04 |
| 1.0 | 2613.05 | 13.89 | 3486.16 | 5.77 |

Table 11: Effect of the temperature coefficient $\tau$ on model performance, using SCIP with a 1,000-second time limit on the IP dataset.

| $\tau$ | Obj $\downarrow$ | $\text{gap}_{abs} \downarrow$ |
|---|---|---|
| 0.05 | 11.47 | 5.57 |
| 0.1 | **11.27** | **5.37** |
| 0.5 | 11.32 | 5.42 |
| 1.0 | 11.51 | 5.61 |

Table 12: Effect of the perturbation ratio on model performance, using SCIP with a 1,000-second time limit on the IP dataset.

| Ratio | Obj $\downarrow$ | gap$_{abs}$ $\downarrow$ |
|-------|------|---------|
| 0.05 | 10.90 | 5.00 |
| 0.10 | 11.30 | 5.40 |
| 0.15 | **10.81** | **4.91** |
| 0.20 | 11.95 | 6.05 |

## H  PERFORMANCE EVALUATION ON WA BENCHMARK

The Workload Appointment (WA)Gasse et al. (2022) dataset is another benchmark from the ML4CO competition. Table 13 presents the performance comparison between our approach and various baselines. When using SCIP as the solver, our method achieves a 58.3% improvement over SCIP and outperforms ConPaS by 17.5%. With Gurobi as the solver, our method improves upon Gurobi by 33.3% and surpasses ConPaS by 22.2%. These results consistently demonstrate the superiority of our approach across different solvers.

Table 13: Performance comparison of our approach against baseline methods on the WA benchmark under a 1,000-second time limit.

| | WA (BKS 707.85) | |
|---|---|---|
| | Obj $\downarrow$ | gap$_{abs}$ $\downarrow$ |
| Gurobi | 708.06 | 0.21 |
| PaS+Gurobi | 708.06 | 0.21 |
| ConPaS+Gurobi | 708.03 | 0.18 |
| Ours+Gurobi | 707.99 | 0.14 |
| SCIP | 710.68 | 2.83 |
| PaS+SCIP | 709.37 | 1.52 |
| ConPaS+SCIP | 709.28 | 1.43 |
| Ours+SCIP | 709.03 | 1.18 |

## I  ADDITIONAL BASELINE

We provide more results by extending our comparison to a recent work, Apollo (Liu et al., 2025). This supplementary experiment offers an updated evaluation against the latest advancement in the field. We reproduce the results of Apollo on the IP (Gasse et al., 2022) and WA (Gasse et al., 2022) datasets using the official code and experimental settings provided by the authors. The outcomes, summarized in Table 14, indicate that our approach outperforms Apollo, yielding a 12.3% improvement on IP dataset and 22% improvement on WA dataset in gap$_{abs}$. These findings highlight the robustness of our method and confirm its competitiveness even when benchmarked against the most recent state-of-the-art approaches. Importantly, our method is complementary to Apollo, suggesting that integrating the two could yield even stronger results.

## J  LLM USAGE

Large Language Models (LLMs) were used solely for translation and language polishing. All research ideas, experiments, and conclusions presented in this paper are entirely the work of the authors.

Table 14: Performance comparison of our approach against Apollo method on the IP benchmark under a 1,000-second time limit.

| | IP (BKS 5.90) | | WA (BKS 707.85) | |
|---|---|---|---|---|
| | Obj $\downarrow$ | gap$_{abs}$ $\downarrow$ | Obj $\downarrow$ | gap$_{abs}$ $\downarrow$ |
| Apollo+Gurobi | 6.55 | 0.65 | 708.03 | 0.18 |
| Ours+Gurobi | 6.47 | 0.57 | 707.99 | 0.14 |

