# OpenReview forum: "HTS-Adapt: A Hybrid Training Strategy with Adaptive Search Region Adjustment for MILPs"
_ICLR.cc/2026/Conference — ICLR 2026 Conference Withdrawn Submission_

### Official Review · Reviewer_gkJE · 2025-10-29

**Soundness:** 3
**Presentation:** 2
**Contribution:** 2
**Rating:** 4
**Confidence:** 4

**Summary:**

This paper proposes HTS-Adapt, an approach that integrates two innovative techniques to enhance the predict-and-search pipeline. For the prediction stage, a hybrid loss combining BCE and contrastive loss is introduced to better identify high-quality solutions. For the search stage, an IIS-based procedure advances the PaS framework by dynamically adjusting the trust region. Experimental results demonstrate the superiority of HTS-Adapt over both the standard PaS and contrastive PaS methods.

**Strengths:**

1,  The hybrid training strategy is insightful. It narrows the scope of "bad" variables via contrastive loss, while directly applying BCE loss to "good" variables—which are less likely to cause infeasibility—is both efficient and effective.
2. The experimental design is sound. The inclusion of solution trajectory visualizations and ablation studies enables a more thorough analysis of the results.

**Weaknesses:**

Baselines: Although Section 2 provides a detailed review of L2O literature, the experimental comparisons are limited to the "PaS → ConPaS → HTS-PaS" pipeline. The authors should consider a broader set of baselines, including learning-based branch-and-cut and learning-based LNS methods.
Novelty: While the HTS component is well-designed, it essentially combines two existing techniques. The IIS component, on the other hand, appears more rooted in operations research than machine learning, and also seems an employment of existing approaches. Thus, the novelty of the paper may be limited.
Clarity on IIS: The description of the IIS procedure is somewhat unclear, and I do not fully understand. For example, in Line 4 of Algorithm 1, what does "fix" entail when $\Delta\neq 0$—hard fixing (as in LNS) or soft fixing (as in local branching)? Additionally, does the second constraint set $C_2$ contain at most one element (i.e., the trust region constraint)?

**Questions:**

1. There is an assumption that "some variables consistently exhibit identical values across high-quality solution sets." This makes sense, but to be more critic, what is the underlying intuition for this phenomenon? Furthermore, what would it imply if this assumption does not hold—would the HTS method reduce to contrastive learning, thereby invalidating the first contribution?
2. Algorithm 1 involves repeatedly solving MILPs within a 1000-second time limit. Since the total time limit in experiments is also set to 1000 seconds, does this imply that the repetition occurs only once?
3. In Figure 2, for MIS and MVC instances, the blue curve (HTS-Adapt) is significantly lower than others from the beginning, suggesting better initial prediction. However, for CA and IP instances, all methods perform comparably in the first 200 seconds, after which HTS-Adapt gradually outperforms the rest. How can this difference be explained? Should it be attributed to better prediction or more effective search?

---

> ### Author Response · Authors · 2025-11-15
>
> We thank the reviewers for their thoughtful feedback and constructive suggestions.
>
> **Weakness 1 (Baselines):**
> Our current experimental setup follows the same benchmark pipeline (PaS → ConPaS) adopted by recent representative works such as **ConPaS** and **Apollo (ICLR 2025)**. We note that learning-based branch-and-cut methods (e.g., for node or variable selection, or cut generation) belong to a different research line from end-to-end neighborhood learning frameworks . Therefore, most prior works in this domain focus on comparisons within the same category of methods to ensure fairness and consistency. In addition to PaS and ConPaS, we have also included results for **Apollo** in Appendix H to provide a broader comparison within this family of approaches.
>
> ---
>
> **Weakness 2 (Novelty):**
> Previous work primarily adopts the loss function proposed by PaS, which is based on cross-entropy. ConPaS, on the other hand, introduce contrastive learning in this domain. Each of the two methods demonstrates advantages on different datasets. By improving and combining these two types of loss functions, our HTS method is able to achieve better performance across various datasets. The IIS component is designed to address the issue of generating infeasible solutions. While previous approaches often attempt to mitigate this by enlarging the search space, our proposed IIS mechanism provides a hard guarantee that complements existing solutions to this situation.
>
> ---
>
> **Weakness 3 (Clarity on IIS):**
> In Line 4, “fix” refers to soft fixing. Specifically, for each variable selected, we introduce an additional constraint together with an auxiliary deviation variable that measures how far the variable is allowed to move away from its predicted value. The sum of all such deviations is bounded by Δ, which effectively restricts the search to a $\delta$-neighborhood.The second constraint set $C_2$ contains **multiple constraints** (typically greater than 20), rather than a single one. These constraints are used to limit the predicted variables within small intervals (e.g., near 0 or 1). If the resulting MILP becomes infeasible, it indicates that a significant portion of the predicted variables were incorrect.
>
> ---
>
> **Question 1:**
> The intuition behind this assumption comes from the strong structural properties of real-world MILPs. Practical instances are not arbitrary: constraint couplings, sparsity patterns, dominance relations, and bottleneck resources often restrict the feasible region in a way that forces certain variables to take the same value across nearly all high-quality solutions. In other words, these variables are not consistent by chance, but because the underlying problem structure makes alternative assignments systematically suboptimal or infeasible.
>
> ---
>
> **Question 2:**
> Although we set a 1000-second time limit for each solver call in Algorithm 1, the process does not necessarily consume the full time. If the MILP is infeasible—which we detect efficiently by checking the infeasibility of the corresponding relaxation—the solver terminates quickly and immediately proceeds to IIS computation. Therefore, repeated iterations can occur within the overall 1000-second experimental limit.
>
> ---
>
> **Question 3:**
> A lower initial gap does not always indicate better variable predictions. Our framework predicts only a subset of variables, leaving the solver to explore the remaining search space. Because the solver uses branch-and-bound, the predictions effectively restrict the search neighborhood, which can influence node selection, variable branching, and the overall search trajectory. Consequently, the same solver may reach different gaps depending on which variables are predicted and how the neighborhood is defined, even if the predictions themselves are accurate.

---

### Official Review · Reviewer_BzGt · 2025-10-31

**Soundness:** 2
**Presentation:** 1
**Contribution:** 2
**Rating:** 4
**Confidence:** 2

**Summary:**

To efficiently solve mixed integer linear programming (MILP), machine learning techniques are used to predict partial solutions, but often suffer from inaccurate and infeasible predictions. Therefore, this work extends the current Contrastive Predictand-Search (ConPaS) framework by two-fold. First, introducing a Hybrid Training Strategy (HTS) to achieve more accurate predictions. Second, proposing an Adaptive Search Region Adjustment mechanism (Adapt) to ensure feasibility and reduce computational overhead.

**Strengths:**

The experiments are sufficient, and the proposed method performs well.

**Weaknesses:**

1. The formatting needs improvement. For example, many citations are missing brackets, which makes the paper difficult to read. The plots in Figures 2 and 3 are too small, while the legends are too large. In addition, the tables should be resized, and the placement of Table 1 is odd—it is introduced in Section 5 but appears at the beginning of Section 4.
2. The preliminaries are insufficient. The work is largely based on ConPaS, but the paper only introduces PaS; the introduction and explanation of ConPaS are lacking.
3. There are several typos. For example, the transpose symbol in Equations (1) and (3). Also, should the c in Equation (2) be bolded? It is inconsistent with Equations (1) and (3).

**Questions:**

1. Why were only MIS and MVC chosen to evaluate generalization?
2. The experiments use SCIP and Gurobi. Given that CPLEX is also a popular MILP solver, why wasn’t CPLEX included?

---

> ### Author Response · Authors · 2025-11-15
>
> We thank the reviewers for their thoughtful feedback and constructive suggestions.
>
> **Weakness 1:**
> We thank the reviewer for pointing this out. These formatting issues do not affect the technical content, and we will address them in the revised manuscript. Specifically, we will correct the citation brackets, adjust the proportions of the figure legends, and move Table 1 to the appendix.
>
> ---
>
> **Weakness 2:**
> We agree that adding a brief explanation of ConPaS improves clarity. We already mention it in the Introduction, and we have added a description in the Preliminaries in the revised manuscript.
>
> ---
>
> **Weakness 3:** We have corrected the transpose symbols in Equations (1) and (3), and We have also unified the formatting of c.
>
> ---
>
> **Question 1:** MIS and MVC are synthetic datasets, which allows us to easily generate larger sets of similar instances for generalization experiments. In contrast, datasets like IP are real-world instances and do not have a straightforward procedure to generate additional similar samples. Therefore, we focus on MIS and MVC when evaluating generalization, while using IP for performance evaluation on real-world data.
>
> ---
>
> **Question 2** In the literature on end-to-end learning–enhanced MILP solvers (e.g., PaS and ConPaS), CPLEX is seldom adopted as an evaluation backend. Following these established experimental protocols, we focused on SCIP and Gurobi, which have become the de facto standard choices in this research line. For completeness, we additionally conducted a small-scale comparison on CPLEX for the IP dataset. The results are consistent with our main findings:
>
> | Method | obj | gap_abs |
> | --- | --- | --- |
> | **SCIP** | 9.14 | 3.24 |
> | **PaS** | 8.89 | 2.99 |
> | **ConPaS** | 8.98 | 3.08 |
> | **Ours** | 8.86 | 2.96 |
>
> This indicates that our method maintains its performance advantage regardless of the backend solver.

---

### Official Review · Reviewer_DwTz · 2025-10-31

**Soundness:** 3
**Presentation:** 3
**Contribution:** 3
**Rating:** 6
**Confidence:** 3

**Summary:**

This paper targets two difficulties of the “predict-then-search” paradigm for Mixed-Integer Linear Programs: (i) learned predictions are often infeasible, and (ii) the search radius is fixed in advance. The authors propose HTS-Adapt, which contains: 1) Hybrid Training Strategy (HTS): supervised cross-entropy loss for “stable” variables that rarely change in near-optimal solutions, and contrastive InfoNCE loss for “volatile” variables; 2) Adaptive Search Region Adjustment (Adapt): whenever the predicted partial assignment is infeasible, an Irreducible Infeasible Subsystem (IIS) is computed to identify the culprit variables and their domains are enlarged selectively instead of naïvely expanding the whole trust region.
Experiments on four classic combinatorial benchmarks (MIS, MVC, CA, IP) show that coupling HTS-Adapt with SCIP reduces the average primal gap by more than 50 % compared with SCIP alone and with previous PaS/ConPaS baselines, while predicting a larger fraction of variables without degrading feasibility.

**Strengths:**

1) Clearly identifies the feasibility and static-radius issues of prior PaS methods.
2) Novel combination of cross-entropy and contrastive learning tailored to variable behaviour, together with targeted expansion of the search region via IIS.
3) Comprehensive empirical evaluation across four datasets; code and hyper-parameters are promised to be released.

**Weaknesses:**

1) Only SCIP is used as the back-end solver; no comparison with Gurobi or CPLEX to demonstrate solver-agnostic benefits.
2) Contrastive component uses plain InfoNCE; more advanced graph-contrastive losses are not explored.
3) No runtime breakdown (prediction / IIS / solver) is given, so the computational overhead of Adapt is unclear.
4) Missing theoretical analysis, e.g., worst-case number of IIS calls needed to regain feasibility.

**Questions:**

1) How much latency does the IIS computation introduce at each node, and is there a lightweight approximate IIS routine for large instances?
2) The threshold for “stable” variables is empirical—does it remain valid across problem types, and could it be learned automatically?
3) When scaling to 10^6 variables, can the GNN still fit in GPU memory, and does the IIS routine remain tractable?

---

> ### Author Response · Authors · 2025-11-15
>
> We thank the reviewers for their thoughtful feedback and constructive suggestions.
>
> **Weakness 1**  The results using **Gurobi** as the back-end solver are provided in **Appendix Table 9** and **Figure 3**, summarized below:
>
> |     | CA (BKS 117223.59) |     | IP (BKS 5.90) |     | MIS (BKS 2628.05) |     | MVC (BKS 3480.39) |     |
> | --- | --- | --- | --- | --- | --- | --- | --- | --- |
> |     | Obj ↓ | gap_abs ↓ | Obj ↓ | gap_abs ↓ | Obj ↑ | gap_abs ↓ | Obj ↓ | gap_abs ↓ |
> | Gurobi | 116576.63 | 646.96 | 6.65 | 0.75 | 2563.70 | 63.18 | 3514.79 | 34.40 |
> | PaS | 116780.64 | 442.95 | 6.60 | 0.70 | 2622.52 | 4.36 | 3481.42 | 1.03 |
> | ConPaS | 116665.31 | 558.28 | 6.57 | 0.67 | 2626.27 | 1.78 | 3481.25 | 0.86 |
> | Ours | 116812.32 | 411.27 | 6.47 | 0.57 | 2626.96 | 1.09 | 3480.64 | 0.25 |
>
> It can be observed that our method outperforms the baseline consistently across four datasets using Gurobi. The results further demonstrate the effectiveness of our method.
>
> ---
>
> **Weakness 2** To isolate and fairly evaluate the contribution of our proposed mechanism, we adopt the same InfoNCE loss used in ConPaS.
>
> ---
>
>  **Weakness 3** The runtime breakdown IIS is provided in **Appendix Table 7**, summarized below:
>
> **IIS Computation Time Statistics**
>
> | Dataset | avg (s) | max (s) | ratio (%) |
> | --- | --- | --- | --- |
> | IP  | 0.27 | 0.29 | 0.03 |
> | MIS | 6.48 | 7.33 | 0.65 |
> | MVC | 4.75 | 5.06 | 0.48 |
> | CA  | 1.33 | 1.45 | 0.13 |
>
> The results show that the IIS computation introduces **negligible overhead**, accounting for **less than 1%** of the total solving time across all datasets. This demonstrates that the *Adapt* module is lightweight and practical for real-world use.
>
> And the prediction stage is also **extremely fast** (typically below 1s) and show as follow:
>
> | Dataset | Prediction Time (s) |
> | --- | --- |
> | MIS | 0.39 |
> | MVC | 0.40 |
> | CA  | 0.38 |
> | IP  | 0.36 |
>
> ---
>
> **Weakness 4** A formal theoretical upper bound on the worst-case number of IIS calls is simply the total number of constraints. Each IIS extraction identifies at least one constraint that conflicts with the predicted assignment, and our update step ensures that this constraint will no longer cause infeasibility in subsequent iterations. Thus, the procedure can terminate after at most ∣C∣ calls in the worst case. Similar to other solver-integrated learning methods, our focus is on empirical efficiency rather than deriving strict worst-case guarantees. In practice, the number of IIS calls is **very small** (typically less than 3 per instance), and the total IIS overhead is **below 1%** of the solving time (see Appendix Table 7). We will clarify this point and discuss potential directions for theoretical analysis in future work.
>
> ---
>
> **Question 1**  In our setting, IIS computation introduces very limited latency, as also evidenced in our analysis for W3. Since IIS is triggered only when the predicted assignment is infeasible and the violated subsystem is typically small, the extraction is fast and adds negligible overhead.
>
> Regarding lightweight approximations, our method does not rely on any specific IIS algorithm. In principle, any fast routine that identifies a small set of constraints responsible for infeasibility can be used in place of the full IIS computation. Designing or integrating such approximate routines for very large instances is orthogonal to our main contribution and is a promising direction for future work.
>
> ---
>
> **Question 2** The reviewer refers to a “threshold,” but our method does not use any tunable threshold. In our design, a variable is marked as *stable* only when **all** perturbed instances yield the **same assignment** (i.e., 100% agreement). This is a deliberately strict and deterministic criterion rather than a hyperparameter.
>
> The intuition behind this rule is supported by the structural properties of real-world MILPs. Practical instances are not arbitrary: constraint couplings, sparsity patterns, dominance relations, and bottleneck resources often restrict the feasible region such that certain variables consistently take the same value across nearly all high-quality solutions. As a consequence, requiring exact agreement across perturbations is both robust and problem-agnostic.
>
> ---
>
> **Question 3** When scaling to 10^6 variables, the GNN cannot fit into GPU memory, and the IIS routine may become computationally expensive. We will add this limitation in the paper. However, it is worth noting that many instances in the MIPLIB2017 dataset have a similar scale to those used in our experiments, thus our results remain representative.

---

### Official Review · Reviewer_J2cw · 2025-11-02

**Soundness:** 3
**Presentation:** 2
**Contribution:** 2
**Rating:** 4
**Confidence:** 4

**Summary:**

The paper improves the predict-and-search framework for solving mixed integer linear programmings (MILP). It mainly addresses two issues, namely prediction accuracy and static search regions that might be missing feasible solutions or unnecessarily large for the search space. To address the first issue, it uses a hybrid training strategy that it separates treatment for stable and unstable variables. Stable variables have identical values across different solutions and are trained with cross-entropy loss for precision. Unstable variable are trained with contrastive loss. This is a hybrid combination of two previous works (PaS and ConPaS). Secondly, it employs adaptive search region adjustment for expanding the search space only when infeasibility arises, guided by irreducible infeasible subsystem (IIS). Experimental results show good performance over four MILP benchmark instances, where the proposed method is better than the baselines in terms of primal solution quality and primal gap, and also faster convergence to good solutions.

**Strengths:**

1. The paper has two main contributions, one is to use the hybrid loss based on variable types for training, and the other is using adaptive search region adjustment.
    2. Experimental results show that the new method outperforms several strong baseline such as PaS and ConPaS in terms of objective values and primal gaps.

**Weaknesses:**

1. The contributions are incremental  It is ok to be incremental if the methods work well with good insights provided either theoretically or empirically. However, the paper mainly describes the methods and shows that it works, but without good explanation or analysis. For example, you could show the precisions of your prediction of the stable/unstable variables compared to PaS and ConPaS
2.  The value additivity of each of the two contributions are not clear. I was looking for a more thorough ablation study (on more MILP problems) to understand the value of each components.
3. The clarity of Section 4.2 is bad. It hurts the overall clarity of the paper since this part is one of the main contributions. I wasn’t able to understand how C1, C2 are computed and how r_max is chosen. Can you provide an algorithmic description of line 11 in Algorithm 1?

**Questions:**

1. In table 4, Adapt-only has smaller values for the best k0, k1 and delta than HTS-only, how do you explain this? How are these parameters determined?
2. How sensitive is your method to parameter r_max? How do you determine its value?

---

> ### Author Response · Authors · 2025-11-15
>
> We thank the reviewer for the feedback.
>
> **Weakness 1 (Incremental contribution):** We would like to emphasize that HTS focuses on capturing the mapping between MILP instances and high-quality solutions, rather than solely predicting stable variables. For stable binary variables, we use label-based learning with cross-entropy loss to capture persistent patterns, while for variables with variation, we employ contrastive learning (InfoNCE) to model relationships among solutions. By effectively combining the strengths of PaS and ConPaS, HTS leverages both types of structure, guiding the solver toward higher-quality solutions.
>
> ---
>
> **Weakness 2 (Value additivity / Ablation):**
> We conducted an additional ablation study on the loss weight $\alpha$ in the **MIS** and **MVC** datasets to evaluate the contribution of the HTS component. The results are shown below:
>
> **Effect of α:**
>
> | α   | MIS |     | MVC |     |
> | --- | --- | --- | --- | --- |
> |     | Obj | gap_abs | Obj | gap_abs |
> | 0.0 | 2618.50 | 8.38 | 3486.21 | 5.82 |
> | 0.2 | **2619.20** | **7.68** | **3485.14** | **4.75** |
> | 0.4 | 2618.99 | 7.89 | 3485.70 | 5.31 |
> | 0.6 | 2618.76 | 8.12 | 3485.23 | 4.84 |
> | 0.8 | 2618.54 | 8.34 | 3485.43 | 5.04 |
> | 1.0 | 2613.05 | 13.89 | 3486.16 | 5.77 |
>
> This study indicates that a moderate $\alpha$ (e.g., 0.2) achieves the best trade-off, validating the necessity of hybrid training.
>
> We also conducted the same ablation experiments on an additional dataset, MVC. The results are as follows:
>
> |     | Obj | gap_abs |
> | --- | --- | --- |
> | ConPaS | 3487.26 | 6.87 |
> | HTS-only | 3485.83 | 5.44 |
> | Adapt-only | 3486.22 | 5.83 |
> | Ours | **3485.70** | **5.31** |
>
> ---
>
> **Weakness 3 (Clarity of Section 4.2):**
> We appreciate the reviewer’s feedback. The parameter $r_{max}$ is chosen empirically based on validation results; Appendix Table 8 reports its influence on performance. The results are as follows:
>
> | r_max | -1  | 40  | 50  | 60  | 100 | 200 |
> | --- | --- | --- | --- | --- | --- | --- |
> | Obj ↓ | 3486.83 | 3486.83 | 3486.83 | 3486.83 | 3487.17 | 3488.33 |
>
> Specifically, we observe that relatively small $r_{max}$values yield more stable and better results, while excessively large ones degrade performance.
> In our method, C1​ and C2​ are obtained via the Gurobi API, which, like other modern MILP solvers (e.g., CPLEX, MindOpt), computes **Irreducible Infeasible Subsystems (IIS)** using standard techniques such as **conflict refinement** or **Farkas-based reduction** to identify minimal sets of inconsistent constraints.
>
> We summarize the key steps for computing an IIS of a given infeasible MILP instance:
>
> 1. Initialization: Start with the full set of constraints of the infeasible MILP as the initial IIS candidate.
>
> 2. Iterative Reduction: Repeatedly examine each constraint in the candidate set:
>
>   - Temporarily remove the constraint and check feasibility.
>
>   - If removing the constraint makes the MILP feasible, it is necessary and should be kept.
>
>   - Otherwise, it is redundant and can be removed.
>
> 3. Termination: Repeat the reduction until no further constraints can be removed.
>
> 4. Output: The remaining constraints form a minimal IIS, representing the smallest subset of constraints that are mutually infeasible.
>
>
> **Pseudocode :**
>
> ```
> Algorithm: IIS Computation for MILP**
>
> Input: Infeasible MILP instance `M` with constraints `C`
> Output: IIS `I ⊆ C`
>
> 1. Initialize IIS candidate `I ← C`
> 2. Set improved flag `γ ← true`
>
> 3. While `γ` is true:
>    1. `γ ← false`
>    2. For each constraint `c` in `I`:
>       1. Temporarily remove `c` from `I`
>       2. Solve MILP with remaining constraints `I \ {c}`
>       3. If MILP is feasible:
>          - `c` is necessary → add it back to `I`
>       4. Else:
>          - `c` is redundant → permanently remove it
>          - `γ ← true`
>
> 4. Return IIS `I`
> ```
>
> ---
>
> **Question 1:**
> For each benchmark, the parameters were tuned to obtain better solutions. Specifically, following **PaS** and **ConPaS**, we fixed $\Delta$ to {0, 5, 10, 20, 50, 100} and varied $k_0$ and $k_1$ from 0% to 50% of the binary variables (in 10% increments). The best values were selected based on validation performance. Since *Adapt-only* was built upon **ConPaS**, its optimal parameters are consistent with those used in ConPaS (i.e., ($k_0$ = 800, $k_1$ = 400, $\Delta$ = 10).
>
> ---
>
> **Question 2:**
> Appendix Table 8 reports the sensitivity of $r_{max}$. We found that smaller values of $r_{max}$ generally produce better and more stable performance, while larger values tend to degrade results. This choice is based on experimental results.

---

### Note · Authors · 2026-01-16

I have read and agree with the venue's withdrawal policy on behalf of myself and my co-authors.